# Exploring Generalization in Deep Learning

**Behnam Neyshabur, Srinadh Bhojanapalli, David McAllester, Nathan Srebro**
Toyota Technological Institute at Chicago
`{bneyshabur, srinadh, mcallester, nati}@ttic.edu`

## Abstract

With a goal of understanding what drives generalization in deep networks, we consider several recently suggested explanations, including norm-based control, sharpness and robustness. We study how these measures can ensure generalization, highlighting the importance of scale normalization, and making a connection between sharpness and PAC-Bayes theory. We then investigate how well the measures explain different observed phenomena.

## 1 Introduction

Learning with deep neural networks has enjoyed huge empirical success in recent years across a wide variety of tasks. Despite being a complex, non-convex optimization problem, simple methods such as stochastic gradient descent (SGD) are able to recover good solutions that minimize the training error. More surprisingly, the networks learned this way exhibit good generalization behavior, even when the number of parameters is significantly larger than the amount of training data [20, 30].

In such an over parametrized setting, the objective has multiple global minima, all minimize the training error, but many of them do not generalize well. Hence, just minimizing the training error is not sufficient for learning: picking the wrong global minima can lead to bad generalization behavior. In such situations, generalization behavior depends implicitly on the algorithm used to minimize the training error. Different algorithmic choices for optimization such as the initialization, update rules, learning rate, and stopping condition, will lead to different global minima with different generalization behavior [7, 12, 18]. For example, Neyshabur et al. [18] introduced Path-SGD, an optimization algorithm that is invariant to rescaling of weights, and showed better generalization behavior over SGD for both feedforward and recurrent neural networks [18, 22]. Keskar et al. [12] noticed that the solutions found by stochastic gradient descent with large batch sizes generalizes worse than the one found with smaller batch sizes, and Hardt et al. [10] discuss how stochastic gradient descent ensures uniform stability, thereby helping generalization for convex objectives.

What is the bias introduced by these algorithmic choices for neural networks? What ensures generalization in neural networks? What is the relevant notion of complexity or capacity control?

As mentioned above, simply accounting for complexity in terms of the number of parameters, or any measure which is uniform across all functions representable by a given architecture, is not sufficient to explain the generalization ability of neural networks trained in practice. For linear models, norms and margin-based measures, and not the number of parameters, are commonly used for capacity control [5, 9, 25]. Also norms such as the trace norm and max norm are considered as sensible inductive biases in matrix factorization and are often more appropriate than parameter-counting measures such as the rank [27, 28]. In a similar spirit, Bartlett [3], Neyshabur et al. [20] and in parallel to this work, Bartlett et al. [2] suggested different norms of network parameters to measure the capacity of neural networks. In a different line of work, Keskar et al. [12] suggested "sharpness" (robustness of the training error to perturbations in the parameters) as a complexity measure for neural networks. Others, including Langford and Caruana [13] and more recently Dziugaite and Roy [8], propose a PAC-Bayes analysis.

What makes a complexity measure appropriate for explaining generalization in deep learning? First, an appropriate complexity measure must be sufficient in ensuring generalization. Second, networks learned in practice should be of low complexity under this measure. This can happen if our optimization algorithms bias us toward lower complexity models under this measure *and* it is possible to capture real data using networks of low complexity. In particular, the complexity measure should help explain several recently observed empirical phenomena that are not explained by a uniform notion of complexity:

- It is possible to obtain zero training error on random labels using the same architecture for which training with real labels leads to good generalization [30]. We would expect the networks learned using real labels (and which generalizes well) to have much lower complexity, under the suggested measure, than those learned using random labels (and which obviously do not generalize well).

- Increasing the number of hidden units, thereby increasing the number of parameters, can lead to a decrease in generalization error even when the training error does not decrease [20]. We would expect to see the complexity measure decrease as we increase the number of hidden units.

- When training the same architecture, with the same training set, using two different optimization methods (or different algorithmic or parameter choices), one method results in better generalization even though both lead to zero training error [18, 12]. We would expect to see a correlation between the complexity measure and generalization ability among zero-training error models.

In this paper we examine complexity measures that have recently been suggested, or could be considered, in explaining generalization in deep learning. We evaluate the measures based on their ability to theoretically guarantee generalization, and their empirical ability to explain the above phenomena. Studying how each measure can guarantee generalization also let us better understand how it should be computed and compared in order to explain the empirical phenomena.

We investigate complexity measures including norms, robustness and sharpness of the network. We emphasize in our theoretical and empirical study the importance of relating the scale of the parameters and the scale of the output of the network, e.g. by relating norm and margin. In this light, we discuss how sharpness by itself is not sufficient for ensuring generalization, but can be combined, through PAC-Bayes analysis, with the norm of the weights to obtain an appropriate complexity measure. The role of sharpness in PAC-Bayesian analysis of neural networks was also recently noted by Dziugaite and Roy [8], who used numerical techniques to numerically optimize the overall PAC-Bayes bound—here we emphasize the distinct role of sharpness as a balance for norm.

**Notation**

Let $f_{\mathbf{w}}(\mathbf{x})$ be the function computed by a $d$ layer feed-forward network with parameters $\mathbf{w}$ and Rectified Linear Unit (ReLU) activations, $f_{\mathbf{w}}(\mathbf{x}) = W_d\,\phi(W_{d-1}\,\phi(....\phi(W_1\mathbf{x})))$ where $\phi(z) = \max\{0, z\}$. Let $h_i$ be the number of nodes in layer $i$, with $h_0 = n$. Therefore, for any layer $i$, we have $W_i \in R^{h_i \times h_{i-1}}$. Given any input $x$, the loss of the prediction by the function $f_{\mathbf{w}}$ is then given by $\ell(\mathbf{w}, \mathbf{x})$. We also denote by $L(\mathbf{w})$ the expected loss and by $\widehat{L}(\mathbf{w})$ the empirical loss over the training set. For any integer $k$, $[k]$ denotes the set $\{1, 2, \cdots, k\}$. Finally, $\|.\|_F$, $\|.\|_2$, $\|.\|_1$, $\|.\|_\infty$ denote Frobenius norm, the spectral norm, element-wise $\ell_1$-norm and element-wise $\ell_\infty$ norm respectively.

## 2   Generalization and Capacity Control in Deep Learning

In this section, we discuss complexity measures that have been suggested, or could be used for capacity control in neural networks. We discuss advantages and weaknesses of each of these complexity measures and examine their abilities to explain the observed generalization phenomena in deep learning.

We consider the statistical *capacity* of a model class in terms of the number of examples required to ensure *generalization*, i.e. that the population (or test error) is close to the training error, even when minimizing the training error. This also roughly corresponds to the maximum number of examples on which one can obtain small training error even with random labels.

Given a model class $\mathcal{H}$, such as all the functions representable by some feedforward or convolutional networks, one can consider the capacity of the entire class $\mathcal{H}$—this corresponds to learning with a uniform "prior" or notion of complexity over all models in the class. Alternatively, we can also consider some *complexity measure*, which we take as a mapping that assigns a non-negative number to every hypothesis in the class - $\mathcal{M} : \{\mathcal{H}, S\} \to \mathbb{R}^+$, where $S$ is the training set. It is then sufficient to consider the capacity of the restricted class $\mathcal{H}_{\mathcal{M},\alpha} = \{h : h \in \mathcal{H}, \mathcal{M}(h) \leq \alpha\}$ for a given $\alpha \geq 0$. One can then ensure generalization of a learned hypothesis $h$ in terms of the capacity of $\mathcal{H}_{\mathcal{M},\mathcal{M}(h)}$. Having a good hypothesis with low complexity, and being biased toward low complexity (in terms of $\mathcal{M}$) can then be sufficient for learning, even if the capacity of the entire $\mathcal{H}$ is high. And if we are indeed relying on $\mathcal{M}$ for ensuring generalization (and in particular, biasing toward models with lower complexity under $\mathcal{M}$), we would expect a learned $h$ with lower value of $\mathcal{M}(h)$ to generalize better. For some of the measures discussed, we allow $\mathcal{M}$ to depend also on the training set. If this is done carefully, we can still ensure generalization for the restricted class $\mathcal{H}_{\mathcal{M},\alpha}$.

We will consider several possible complexity measures. For each candidate measure, we first investigate whether it is sufficient for generalization, and analyze the capacity of $\mathcal{H}_{\mathcal{M},\alpha}$. Understanding the capacity corresponding to different complexity measures also allows us to relate between different measures and provides guidance as to what and how we should measure: From the above discussion, it is clear that any monotone transformation of a complexity measures leads to an equivalent notion of complexity. Furthermore, complexity is meaningful only in the context of a specific hypothesis class $\mathcal{H}$, e.g. specific architecture or network size. The capacity, as we consider it (in units of sample complexity), provides a yardstick by which to measure complexity (we should be clear though, that we are vague regarding the scaling of the generalization error itself, and only consider the scaling in terms of complexity and model class, thus we obtain only a very crude yardstick sufficient for investigating trends and relative phenomena, not a quantitative yardstick).

## 2.1 Network Size

For any model, if its parameters have finite precision, its capacity is linear in the total number of parameters. Even without making an assumption on the precision of parameters, the VC dimension of feedforward networks can be bounded in terms of the number of parameters $\dim(\mathbf{w})$[1, 3, 6, 23]. In particular, Bartlett [4] and Harvey et al. [11], following Bartlett et al. [6], give the following tight (up to logarithmic factors) bound on the VC dimension and hence capacity of feedforward networks with ReLU activations:

$$\text{VC-dim} = \tilde{O}(d * \dim(\mathbf{w})) \tag{1}$$

In the over-parametrized settings, where the number of parameters is more than the number of samples, complexity measures that depend on the total number of parameters are too weak and cannot explain the generalization behavior. Neural networks used in practice often have significantly more parameters than samples, and indeed can perfectly fit even random labels, obviously without generalizing [30]. Moreover, measuring complexity in terms of number of parameters cannot explain the reduction in generalization error as the number of hidden units increase [20] (see also Figure 4).

## 2.2 Norms and Margins

Capacity of linear predictors can be controlled independent of the number of parameters, e.g. through regularization of its $\ell_2$ norm. Similar norm based complexity measures have also been established for feedforward neural networks with ReLU activations. For example, capacity can be bounded based on the $\ell_1$ norm of the weights of hidden units in each layer, and is proportional to $\prod_{i=1}^{d} \|W_i\|_{1,\infty}^2$, where $\|W_i\|_{1,\infty}$ is the maximum over hidden units in layer $i$ of the $\ell_1$ norm of incoming weights to the hidden unit [5]. More generally Neyshabur et al. [19] considered group norms $\ell_{p,q}$ corresponding to $\ell_q$ norm over hidden units of $\ell_p$ norm of incoming weights to the hidden unit. This includes $\ell_{2,2}$ which is equivalent to the Frobenius norm where the capacity of the network is proportional to $\prod_{i=1}^{d} \|W_i\|_F^2$. They further motivated a complexity measure that is invariant to node-wise rescaling reparametrization [1], suggesting $\ell_p$ path norms which is the minimum over all node-wise rescalings of $\prod_{i=1}^{d} \|W_i\|_{p,\infty}$ and is equal to $\ell_p$ norm of a vector with coordinates each of which is the product

of weights along a path from an input node to an output node in the network. While preparing this manuscript, we became aware of parallel work Bartlett et al. [2] that proves generalization bounds with capacity is proportional to $\prod_{i=1}^{d} \|W_i\|_2^2 \left( \sum_{j=1}^{d} \left( \|W_j\|_1 / \|W_j\|_2 \right)^{2/3} \right)^3$.

Capacity control in terms of norm, when using a zero/one loss (i.e. counting errors) requires us in addition to account for scaling of the output of the neural networks, as the loss is insensitive to this scaling but the norm only makes sense in the context of such scaling. For example, dividing all the weights by the same number will scale down the output of the network but does not change the $0/1$ loss, and hence it is possible to get a network with arbitrary small norm and the same $0/1$ loss. Using a scale sensitive losses, such as the cross entropy loss, does address this issue (if the outputs are scaled down toward zero, the loss becomes trivially bad), and one can obtain generalization guarantees in terms of norm and the cross entropy loss.

However, we should be careful when comparing the norms of different models learned by minimizing the cross entropy loss, in particular when the training error goes to zero. When the training error goes to zero, in order to push the cross entropy loss (or any other positive loss that diminish at infinity) to zero, the outputs of the network must go to infinity, and thus the norm of the weights (under any norm) should also go to infinity. This means that minimizing the cross entropy loss will drive the norm toward infinity. In practice, the search is terminated at some finite time, resulting in large, but finite norm. But the value of this norm is mostly an indication of how far the optimization is allowed to progress—using a stricter stopping criteria (or higher allowed number of iterations) would yield higher norm. In particular, comparing the norms of models found using different optimization approaches is meaningless, as they would all go toward infinity.

Instead, to meaningfully compare norms of the network, we should explicitly take into account the scaling of the outputs of the network. One way this can be done, when the training error is indeed zero, is to consider the "margin" of the predictions in addition to the norms of the parameters. We refer to the margin for a single data point $x$ as the difference between the score of the correct label and the maximum score of other labels, i.e.

$$f_{\mathbf{w}}(\mathbf{x})[y_{\text{true}}] - \max_{y \neq y_{\text{true}}} f_{\mathbf{w}}(\mathbf{x})[y] \tag{2}$$

In order to measure scale over an entire training set, one simple approach is to consider the "hard margin", which is the minimum margin among all training points. However, this definition is very sensitive to extreme points as well as to the size of the training set. We consider instead a more robust notion that allows a small portion of data points to violate the margin. For a given training set and small value $\epsilon > 0$, we define the margin $\gamma_{\text{margin}}$ as the lowest value of $\gamma$ such that $\lceil \epsilon m \rceil$ data point have margin lower than $\gamma$ where $m$ is the size of the training set. We found empirically that the qualitative and relative nature of our empirical results is almost unaffected by reasonable choices of $\epsilon$ (e.g. between $0.001$ and $0.1$).

The measures we investigate in this work and their corresponding capacity bounds are as follows [2]:

- $\ell_2$ norm with capacity proportional to $\frac{1}{\gamma_{\text{margin}}^2} \prod_{i=1}^{d} 4 \|W_i\|_F^2$ [19].

- $\ell_1$-path norm with capacity proportional to $\frac{1}{\gamma_{\text{margin}}^2} \left| \sum_{j \in \prod_{k=0}^{d} [h_k]} \left| \prod_{i=1}^{d} 2W_i[j_i, j_{i-1}] \right| \right|^2$ [5, 19].

- $\ell_2$-path norm with capacity proportional to $\frac{1}{\gamma_{\text{margin}}^2} \sum_{j \in \prod_{k=0}^{d} [h_k]} \prod_{i=1}^{d} 4h_i W_i^2[j_i, j_{i-1}]$.

- spectral norm with capacity proportional to $\frac{1}{\gamma_{\text{margin}}^2} \prod_{i=1}^{d} h_i \|W_i\|_2^2$.

where $\prod_{k=0}^{d} [h_k]$ is the Cartesian product over sets $[h_k]$. The above bounds indicate that capacity can be bounded in terms of either $\ell_2$-norm or $\ell_1$-path norm independent of number of parameters. The

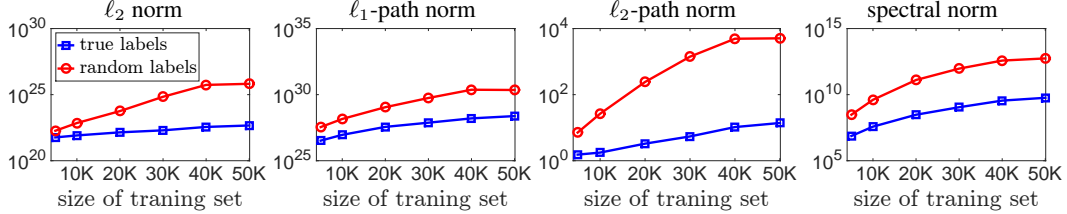

Figure 1: Comparing different complexity measures on a VGG network trained on subsets of CIFAR10 dataset with true (blue line) or random (red line) labels. We plot norm divided by margin to avoid scaling issues (see Section 2), where for each complexity measure, we drop the terms that only depend on depth or number of hidden units; e.g. for $\ell_2$-path norm we plot $\gamma_{\text{margin}}^{-2} \sum_{j \in \prod_{k=0}^{d}[h_k]} \prod_{i=1}^{d} W_i^2[j_i, j_{i-1}]$. We also set the margin over training set $S$ to be $5^{th}$-percentile of the margins of the data points in $S$, i.e. $\text{Prc}_5 \{f_{\mathbf{w}}(x_i)[y_i] - \max_{y \neq y_i} f_{\mathbf{w}}(\mathbf{x})[y]|(x_i, y_i) \in S\}$. In all experiments, the training error of the learned network is zero. The plots indicate that these measures can explain the generalization as the complexity of model learned with random labels is always higher than the one learned with true labels. Moreover, the gap between the complexity of models learned with true and random labels increases as we increase the size of the training set.

$\ell_2$-path norm dependence on the number of hidden units in each layer is unavoidable. However, it is not clear if a bound that only depends on the product of spectral norms is possible.

As an initial empirical investigation of the appropriateness of the different complexity measures, we compared the complexity (under each of the above measures) of models trained on true versus random labels. We would expect to see two phenomena: first, the complexity of models trained on true labels should be substantially lower than those trained on random labels, corresponding to their better generalization ability. Second, when training on random labels, we expect capacity to increase almost linearly with the number of training examples, since every extra example requires new capacity in order to fit it's random label. However, when training on true labels we expect the model to capture the true functional dependence between input and output and thus fitting more training examples should only require small increases in the capacity of the network. The results are reported in Figure 1. We indeed observe a gap between the complexity of models learned on real and random labels for all four norms, with the difference in increase in capacity between true and random labels being most pronounced for the $\ell_2$ norm and $\ell_2$-path norm.

**Lipschitz Continuity and Robustness**  The measures/norms we discussed so far also control the Lipschitz constant of the network with respect to its input. Is the capacity control achieved through the bound on the Lipschitz constant? Is bounding the Lipschitz constant alone enough for generalization? In Appendix A, we show that the current bounds using Lipschitz have exponential dependence to the input dimension and therefore the capacity bounds discussed above are not merely a consequence of bounding the Lipschitz constant.

In Section 3 we present further empirical investigations of the appropriateness of these complexity measures to explain other phenomena.

## 2.3 Sharpness

The notion of sharpness as a generalization measure was recently suggested by Keskar et al. [12] and corresponds to robustness to adversarial perturbations on the parameter space:

$$\zeta_\alpha(\mathbf{w}) = \frac{\max_{|\boldsymbol{\nu}_i| \leq \alpha(|\mathbf{w}_i|+1)} \widehat{L}(f_{\mathbf{w}+\boldsymbol{\nu}}) - \widehat{L}(f_{\mathbf{w}})}{1 + \widehat{L}(f_{\mathbf{w}})} \simeq \max_{|\boldsymbol{\nu}_i| \leq \alpha(|\mathbf{w}_i|+\mathbf{1})} \widehat{L}(f_{\mathbf{w}+\boldsymbol{\nu}}) - \widehat{L}(f_{\mathbf{w}}), \quad (3)$$

where the training error $\widehat{L}(f_{\mathbf{w}})$ is generally very small in the case of neural networks in practice, so we can simply drop it from the denominator without a significant change in the sharpness value.

As we will explain below, sharpness defined this way does *not* capture the generalization behavior. To see this, we first examine whether sharpness can predict the generalization behavior for networks trained on true vs random labels. In the left plot of Figure 2, we plot the sharpness for networks trained on true vs random labels. While sharpness correctly predicts the generalization behavior for

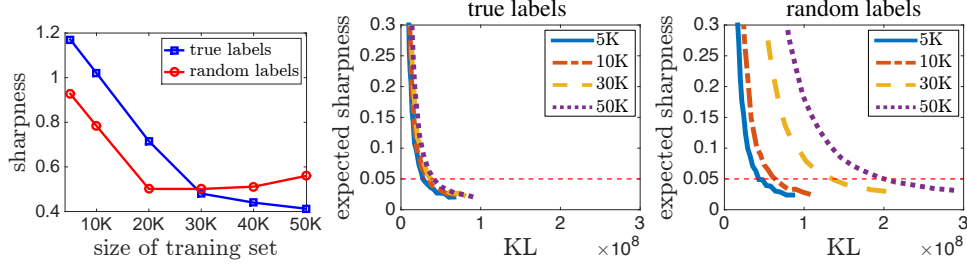

Figure 2: Sharpness and PAC-Bayes measures on a VGG network trained on subsets of CIFAR10 dataset with true or random labels. In the left panel, we plot max sharpness, calculated as suggested by Keskar et al. [12] where the perturbation for parameter $w_i$ has magnitude $5.10^{-4}(|w_i| + 1)$. The middle and right plots show the relationship between expected sharpness and KL divergence in PAC-Bayes bound for true and random labels respectively. For PAC-Bayes plots, each point in the plot correspond to a choice of $\alpha$ where the standard deviation of the perturbation for the parameter $w_i$ is $\alpha(10|w_i| + 1)$. The corresponding $KL$ to each $\alpha$ is weighted $\ell_2$ norm where the weight for each parameter is the inverse of the standard deviation of the perturbation.

bigger networks, for networks of smaller size, those trained on random labels have less sharpness than the ones trained on true labels. Furthermore sharpness defined above depends on the scale of $\mathbf{w}$ and can be artificially increased or decreased by changing the scale of the parameters. Therefore, sharpness alone is not sufficient to control the capacity of the network.

Instead, we advocate viewing a related notion of expected sharpness in the context of the PAC-Bayesian framework. Viewed this way, it becomes clear that sharpness controls only one of two relevant terms, and must be balanced with some other measure such as norm. Together, sharpness and norm do provide capacity control and can explain many of the observed phenomena. This connection between sharpness and the PAC-Bayes framework was also recently noted by Dziugaite and Roy [8].

The PAC-Bayesian framework [16, 17] provides guarantees on the expected error of a randomized predictor (hypothesis), drawn form a distribution denoted $\mathcal{Q}$ and sometimes referred to as a "posterior" (although it need *not* be the Bayesian posterior), that depends on the training data. Let $f_{\mathbf{w}}$ be any predictor (not necessarily a neural network) learned from training data. We consider a distribution $\mathcal{Q}$ over predictors with weights of the form $\mathbf{w} + \boldsymbol{\nu}$, where $\mathbf{w}$ is a single predictor learned from the training set, and $\boldsymbol{\nu}$ is a random variable. Then, given a "prior" distribution $P$ over the hypothesis that is independent of the training data, with probability at least $1 - \delta$ over the draw of the training data, the expected error of $f_{\mathbf{w}+\boldsymbol{\nu}}$ can be bounded as follows [15]:

$$\mathbb{E}_{\boldsymbol{\nu}}[L(f_{\mathbf{w}+\boldsymbol{\nu}})] \leq \mathbb{E}_{\boldsymbol{\nu}}[\widehat{L}(f_{\mathbf{w}+\boldsymbol{\nu}})] + 4\sqrt{\frac{\left(KL\left(\mathbf{w} + \boldsymbol{\nu}\|P\right) + \ln\frac{2m}{\delta}\right)}{m}} \tag{4}$$

Substituting $\mathbb{E}_{\boldsymbol{\nu}}[\widehat{L}(f_{\mathbf{w}+\boldsymbol{\nu}})]$ with $\widehat{L}(f_{\mathbf{w}}) + \left(\mathbb{E}_{\boldsymbol{\nu}}[\widehat{L}(f_{\mathbf{w}+\boldsymbol{\nu}})] - \widehat{L}(f_{\mathbf{w}})\right)$ we can see that the PAC-Bayes bound depends on two quantities - i) the expected sharpness and ii) the Kullback Leibler (KL) divergence to the "prior" $P$. The bound is valid for any distribution measure $P$, any perturbation distribution $\boldsymbol{\nu}$ and any method of choosing $\mathbf{w}$ dependent on the training set. A simple way to instantiate the bound is to set $P$ to be a zero mean, $\sigma^2$ variance Gaussian distribution. Choosing the perturbation $\boldsymbol{\nu}$ to also be a zero mean spherical Gaussian with variance $\sigma^2$ in every direction, yields the following guarantee (w.p. $1 - \delta$ over the training set):

$$\mathbb{E}_{\boldsymbol{\nu}\sim\mathcal{N}(0,\sigma)^n}[L(f_{\mathbf{w}+\boldsymbol{\nu}})] \leq \widehat{L}(f_{\mathbf{w}}) + \underbrace{\mathbb{E}_{\boldsymbol{\nu}\sim\mathcal{N}(0,\sigma)^n}[\widehat{L}(f_{\mathbf{w}+\boldsymbol{\nu}})] - \widehat{L}(f_{\mathbf{w}})}_{\text{expected sharpness}} + 4\sqrt{\frac{1}{m}\left(\underbrace{\frac{\|\mathbf{w}\|_2^2}{2\sigma^2}}_{\text{KL}} + \ln\frac{2m}{\delta}\right)}, \quad (5)$$

Another interesting approach is to set the variance of the perturbation to each parameter with respect to the magnitude of the parameter. For example if $\sigma_i = \alpha|w_i| + \beta$, then the KL term in the above expression changes to $\sum_i \frac{w_i^2}{2\sigma_i^2}$. The above generalization guarantees give a clear way to think about capacity control jointly in terms of both the expected sharpness and the norm, and as we discussed earlier indicates that sharpness by itself cannot control the capacity without considering the scaling. In the above generalization bound, norms and sharpness interact in a direct way depending on $\sigma$,

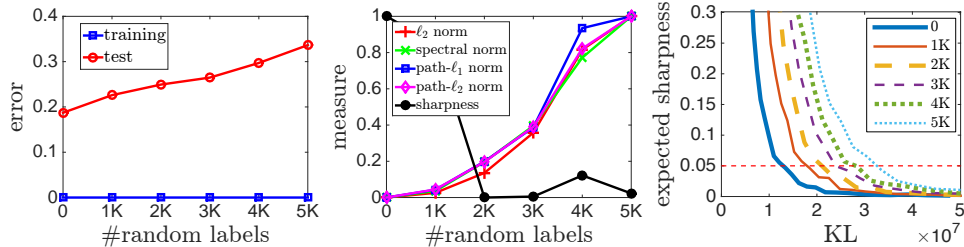

Figure 3: Experiments on global minima with poor generalization. For each experiment, a VGG network is trained on union of a subset of CIFAR10 with size 10000 containing samples with true labels and another subset of CIFAR10 datasets with varying size containing random labels. The learned networks are all global minima for the objective function on the subset with true labels. The left plot indicates the training and test errors based on the size of the set with random labels. The plot in the middle shows change in different measures based on the size of the set with random labels. The plot on the right indicates the relationship between expected sharpness and KL in PAC-bayes for each of the experiments. Measures are calculated as explained in Figures 1 and 2.

as increasing the norm by decreasing $\sigma$ causes decrease in sharpness and vice versa. It is therefore important to find the right balance between the norm and sharpness by choosing $\sigma$ appropriately in order to get a reasonable bound on the capacity.

In our experiments we observe that looking at both these measures jointly indeed makes a better predictor for the generalization error. As discussed earlier, Dziugaite and Roy [8] numerically optimize the overall PAC-Bayes generalization bound over a family of multivariate Gaussian distributions (different choices of perturbations and priors). Since the precise way the sharpness and KL-divergence are combined is not tight, certainly not in (5), nor in the more refined bound used by Dziugaite and Roy [8], we prefer shying away from numerically optimizing the balance between sharpness and the KL-divergence. Instead, we propose using bi-criteria plots, where sharpness and KL-divergence are plotted against each other, as we vary the perturbation variance. For example, in the center and right panels of Figure 2 we show such plots for networks trained on true and random labels respectively. We see that although sharpness by itself is not sufficient for explaining generalization in this setting (as we saw in the left panel), the bi-criteria plots are significantly lower for the true labels. Even more so, the change in the bi-criteria plot as we increase the number of samples is significantly larger with random labels, correctly capturing the required increase in capacity. For example, to get a fixed value of expected sharpness such as $\epsilon = 0.05$, networks trained with random labels require higher norm compared to those trained with true labels. This behavior is in agreement with our earlier discussion, that sharpness is sensitive to scaling of the parameters and is not a capacity control measure as it can be artificially changed by scaling the network. However, combined with the norm, sharpness does seem to provide a capacity measure.

## 3   Empirical Investigation

In this section we investigate the ability of the discussed measures to explain the the generalization phenomenon discussed in the Introduction. We already saw in Figures 1 and 2 that these measures capture the difference in generalization behavior of models trained on true or random labels, including the increase in capacity as the sample size increases, and the difference in this increase between true and random labels.

**Different Global Minima**   Given different global minima of the training loss on the same training set and with the same model class, can these measures indicate which model is going to generalize better? In order to verify this property, we can calculate each measure on several different global minima and see if lower values of the measure imply lower generalization error. In order to find different global minima for the training loss, we design an experiment where we force the optimization methods to converge to different global minima with varying generalization abilities by forming a confusion set that includes samples with random labels. The optimization is done on the loss that includes examples from both the confusion set and the training set. Since deep learning models have very high capacity, the optimization over the union of confusion set and training set generally leads to a point with zero error over both confusion and training sets which thus is a global minima for the

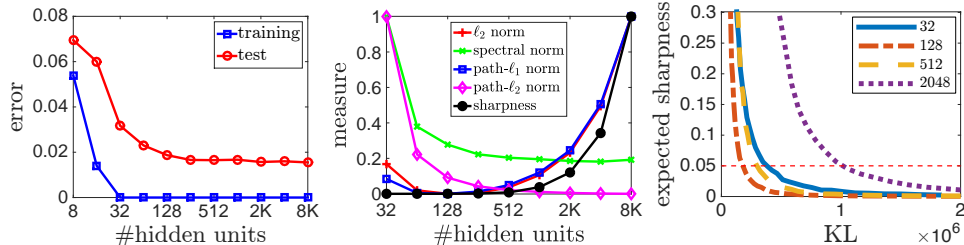

Figure 4: The generalization of two layer perceptron trained on MNIST with varying number of hidden units. The left plot indicates the training and test errors. The test error decreases as the size increases. The middle plot shows measures for each of the trained networks. The plot on the right indicates the relationship between sharpness and KL in PAC-Bayes for each experiment. Measures are calculated as explained in Figures 1 and 2.

training set. We randomly select a subset of CIFAR10 dataset with 10000 data points as the training set and our goal is to find networks that have zero error on this set but different generalization abilities on the test set. In order to do that, we train networks on the union of the training set with fixed size 10000 and confusion sets with varying sizes that consists of CIFAR10 samples with random labels; and we evaluate the learned model on an independent test set. The trained network achieves zero training error but as shown in Figure 3, the test error of the model increases with increasing size of the confusion set. The middle panel of this Figure suggests that the norm of the learned networks can indeed be predictive of their generalization behavior. However, we again observe that sharpness has a poor behavior in these experiments. The right panel of this figure also suggests that PAC-Bayes measure of joint sharpness and KL divergence, has better behavior - for a fixed expected sharpness, networks that have higher generalization error, have higher norms.

**Increasing Network Size** We also repeat the experiments conducted by Neyshabur et al. [20] where a fully connected feedforward network is trained on MNIST dataset with varying number of hidden units and we check the values of different complexity measures on each of the learned networks. The left panel in Figure 4 shows the training and test error for this experiment. While 32 hidden units are enough to fit the training data, we observe that networks with more hidden units generalize better. Since the optimization is done without any explicit regularization, the only possible explanation for this phenomenon is the implicit regularization by the optimization algorithm. Therefore, we expect a sensible complexity measure to decrease beyond 32 hidden units and behave similar to the test error. Different measures are reported for learned networks. The middle panel suggest that all margin/norm based complexity measures decrease for larger networks up to 128 hidden units. For networks with more hidden units, $\ell_2$ norm and $\ell_1$-path norm increase with the size of the network. The middle panel suggest that $\ell_2$-path norm and spectral norm can provide some explanation for this phenomenon. However, as we discussed in Section 2, the actual complexity measure based on $\ell_2$-path norm and spectral norm also depends on the number of hidden units and taking this into account indicates that these measures cannot explain this phenomenon. In Appendix A, we discuss another complexity measure that also depends the spectral norm through Lipschitz continuity or robustness argument. Even though this bound is very loose (exponential in input dimension), it is monotonic with respect to the spectral norm that is reported in the plots. The right panel shows that the joint PAC-Bayes measure decrease for larger networks up to size 128 but fails to explain this generalization behavior for larger networks. This suggests that the measures looked so far are not sufficient to explain all the generalization phenomenon observed in neural networks.

## 4 Conclusion

Learning with deep neural networks displays good generalization behavior in practice, a phenomenon that remains largely unexplained. In this paper we discussed different candidate complexity measures that might explain generalization in neural networks. We outline a concrete methodology for investigating such measures, and report on experiments studying how well the measures explain different phenomena. While there is no clear choice yet, some combination of expected sharpness and norms do seem to capture much of the generalization behavior of neural networks. A major issue still left unresolved is how the choice of optimization algorithm biases such complexity to be low, and what is the precise relationship between optimization and implicit regularization.

## Footnotes

[1]Node-rescaling can be defined as a sequence of reparametrizations, each of which corresponds to multiplying incoming weights and dividing outgoing weights of a hidden unit by a positive scalar $\alpha$. The resulting network computes the same function as the network before the reparametrization.

[2] We have dropped the term that only depends on the norm of the input. The bounds based on $\ell_2$-path norm and spectral norm can be derived directly from the those based on $\ell_1$-path norm and $\ell_2$ norm respectively. Without further conditions on weights, exponential dependence on depth is tight but the $4^d$ dependence might be loose [19]. As we discussed at the beginning of this subsection, in parallel work, Bartlett et al. [2] have improved the spectral bound.

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
