[Supplementary Material]

# A Lipschitz Continuity and Robustness

In order to understand capacity control in terms of Lipschitz continuity, we review here the relevant guarantees. Given an input space $\mathcal{X}$ and metric $\mathcal{M}$, a function $f : \mathcal{X} \to \mathbb{R}$ on a metric space $(\mathcal{X}, \mathcal{M})$ is called a Lipschitz function if there exists a constant $C_{\mathcal{M}}$, such that $|f(x) - f(y)| \leq C_{\mathcal{M}} \mathcal{M}(x, y)$. Luxburg and Bousquet [14] studied the capacity of functions with bounded Lipschitz constant on metric space $(\mathcal{X}, \mathcal{M})$ with a finite diameter $\text{diam}_{\mathcal{M}}(\mathcal{X}) = \sup_{x,y \in X} \mathcal{M}(x, y)$ and showed that the capacity is proportional to $\left( \frac{C_{\mathcal{M}}}{\gamma_{\text{margin}}} \right)^n \text{diam}_{\mathcal{M}}(\mathcal{X})$. This capacity bound is weak as it has an exponential dependence on input size.

Another related approach is through algorithmic robustness as suggested by Xu and Mannor [29]. Given $\epsilon > 0$, the model $f_{\mathbf{w}}$ found by a learning algorithm is $K$ robust if $\mathcal{X}$ can be partitioned into $K$ disjoint sets, denoted as $\{C_i\}_{i=1}^K$, such that for any pair $(\mathbf{x}, y)$ in the training set $\mathbf{s}$ ,[3]

$$\mathbf{x}, \mathbf{z} \in C_i \Rightarrow |\ell(\mathbf{w}, \mathbf{x}) - \ell(\mathbf{w}, \mathbf{z})| \leq \epsilon \tag{6}$$

Xu and Mannor [29] showed the capacity of a model class whose models are $K$-robust scales as $K$. For the model class of functions with bounded Lipschitz $C_{\|.\|}$, $K$ is proportional to $\frac{C_{\|.\|}}{\gamma_{\text{margin}}}$-covering number of the input domain $\mathcal{X}$ under norm $\|.\|$. However, the covering number of the input domain can be exponential in the input dimension and the capacity can still grow as $\left( \frac{C_{\|.\|}}{\gamma_{\text{margin}}} \right)^n$ [4].

Returning to our original question, the $C_{\ell_\infty}$ and $C_{\ell_2}$ Lipschitz constants of the network can be bounded by $\prod_{i=1}^d \|W_i\|_{1,\infty}$ (hence $\ell_1$-path norm) and $\prod_{i=1}^d \|W_i\|_2$, respectively [29, 26]. This will result in a very large capacity bound that scales as $\left( \frac{\prod_{i=1}^d \|W_i\|_2}{\gamma_{\text{margin}}} \right)^n$, which is exponential in both the input dimension and depth of the network. This shows that simply bounding the Lipschitz constant of the network is not enough to get a reasonable capacity control.

# B Experiments Settings

In experiment with different network sizes, we train a two layer perceptron with ReLU activation and varying number of hidden units without Batch Normalization or dropout. In the rest of the experiments, we train a modified version of the VGG architecture [24] with the configuration $2 \times [64, 3, 3, 1]$, $2 \times [128, 3, 3, 1]$, $2 \times [256, 3, 3, 1]$, $2 \times [512, 3, 3, 1]$ where we add Batch Normalization before ReLU activations and apply $2 \times 2$ max-pooling with window size 2 and dropout after each stack. Convolutional layers are followed by $4 \times 4$ average pooling, a fully connected layer with 512 hidden units and finally a linear layer is added for prediction.

In all experiments we train the networks using stochastic gradient descent (SGD) with mini-batch size 64, fixed learning rate 0.01 and momentum 0.9 without weight decay. In all experiments where achieving zero training error is possible, we continue training until the cross-entropy loss is less than $10^{-4}$.

When calculating norms on a network with a Batch Normalization layer, we reparametrize the network to one that represents the exact same function without Batch Normalization as suggested in [21]. In all our figures we plot norm divided by margin to avoid scaling issues (see Section 2), where we set the margin over training set $S$ to be $5^{th}$-percentile of the margins of the data points in $S$, i.e. $\text{Prc}_5 \{ f_{\mathbf{w}}(x_i)[y_i] - \max_{y \neq y_i} f_{\mathbf{w}}(x)[y] | (x_i, y_i) \in S \}$ . We have also investigated other versions of the margin and observed similar behavior to this notion.

We calculate the sharpness, as suggested in [12] - for each parameter $w_i$ we bound the magnitude of perturbation by $\alpha(|w_i| + 1)$ for $\alpha = 5.10^{-4}$. In order to compute the maximum perturbation (maximize the loss), we perform 2000 updates of stochastic gradient ascent starting from the minimum, with mini-batch size 64, fixed step size 0.01 and momentum 0.9.

To compute the expected sharpness, we perturb each parameter $w_i$ of the model with noise generated from Gaussian distribution with zero mean and standard deviation, $\alpha(10 |w_i| + 1)$. The expected

sharpness is average over 1000 random perturbations each of which are averaged over a mini-batch of size 64. We compute the expected sharpness for different choices of $\alpha$. For each value of $\alpha$ the KL divergence can be calculated as $\frac{1}{\alpha^2} \sum_i \left( \frac{w_i}{(10|w_i|+1)} \right)^2$.

## C  Bounding Sharpness

We have discussed margin based and sharpness based complexity measures to understand capacity. We have also discussed how sharpness based complexity measures in combination with norms characterize the generalization behavior under the PAC-Bayes framework. In this section we study the question of what affects the sharpness of neural networks? For the case of linear predictors, sharpness only depends on the norm of the predictor. In contrast, for multilayered networks, interaction between the layers plays a major role and consequently two different networks with the same norm can have drastically different sharpness values. For example, consider a network where some subset of the layers despite having non-zero norm interact weakly with their neighbors, or are almost orthogonal to each other. Such a network will have very high sharpness value compared to a network where the neighboring layers interact strongly.

In this section we establish sufficient conditions to bound the expected sharpness of a feedforward network with ReLU activations. Such conditions serve as a useful guideline in studying what helps an optimization method to converge to less sharp optima. Unlike existing generalization bounds [5, 19, 14, 29, 26], our sharpness based bound does not suffer from exponential dependence on depth.

For a given $\mathbf{x} \in \mathbb{R}^n$, let $D_i^{\mathbf{x},\mathbf{w}}$ denote the diagonal $\{1,0\}$ matrix corresponding to activation in layer $i$. To simplify the presentation we drop the $\mathbf{x}$ superscript and use $D_i$ instead. We can therefore write $f_\mathbf{w}(\mathbf{x}) = W_d \, D_{d-1} \, W_{d-1} \cdots D_1 \, W_1 \, \mathbf{x} = W_d \left( \Pi_{i=1}^{d-1} D_i W_i \right) \mathbf{x}$ where we drop the $\mathbf{x}, \mathbf{w}$ superscript from $D_i^{\mathbf{x},\mathbf{w}}$ and use $D_i$ instead but remember that $D_i$ depends on $\mathbf{x}$ and the parameters $W_j$ for any $j \leq i$.

Now we discuss the conditions that affect the sharpness of a network. As discussed earlier, weak interactions between layers can cause the network to have high sharpness value. Condition $C1$ below prevents such weak interactions (cancellations). A network can also have high sharpness if the changes in the number of activations is exponential in the perturbations to its weights, even for small perturbations. Condition $C2$ avoids such extreme situations on activations. Finally, if a non-active node with large weights becomes active because of the perturbations in lower layers, that can lead to huge changes to the output of the network. Condition $C3$ prevents having such spiky (in magnitude) hidden units. This leads us to the following three conditions, that help in avoiding such pathological cases.

$(C1):$ Given $x$, let $x = W_0$ and $D_0 = I$. Then, for all $0 \leq a < c < b \leq d, \| \left( \Pi_{i=a}^b D_i W_i \right) \|_F \geq \frac{\mu}{\sqrt{h_c}} \| \Pi_{i=c+1}^b D_i W_i \|_F \| \left( \Pi_{i=a}^c D_i W_i \right) \|_F.$

$(C2):$ Given $x$, for any level $k$, $\frac{1}{h_k} \sum_{i \in [h_k]} 1_{W_{k,i} \Pi_{j=1}^{k-1} D_j W_j x \leq \delta} \leq C_2 \delta.$

$(C3):$ For all $i$, $\|W_i\|_{2,\infty}^2 h_i \leq C_3^2 \|D_i W_i\|_F^2.$

Here, $W_{k,i}$ denotes the weights of the $i^{th}$ output node in layer $k$. $\|W_i\|_{2,\infty}$ denotes the maximum $L2$ norm of a hidden unit in layer $i$. Now we state our result on the generalization error of a ReLU network, in terms of average sharpness and its norm. Let $\|x\| = 1$ and $h = \max_{i=1}^d h_i$.

**Theorem 1.** *Let $\boldsymbol{\nu}_i$ be a random $h_i \times h_{i-1}$ matrix with each entry distributed according to $\mathcal{N}(0, \sigma_i^2)$. Then, under the conditions $C1, C2, C3$, with probability $\geq 1 - \delta$,*

$$\mathbb{E}_{\boldsymbol{\nu} \sim \mathcal{N}(0,\sigma)^n} [L(f_{\mathbf{w}+\boldsymbol{\nu}})] - \widehat{L}(f_\mathbf{w}) \leq O \left( \left[ \Pi_{i=1}^d \left(1 + \gamma_i\right) - 1 \right. \right.$$

$$\left. \left. + \Pi_{i=1}^d \left(1 + \gamma_i C_2 C_3 \right) \left( \Pi_{i=1}^d (1 + \gamma_i C_\delta C_2) - 1 \right) \right] C_L \sum_x \frac{\|f_\mathbf{w}(x)\|_F}{m} \right) + \sqrt{\frac{1}{m} \left( \sum_{i=1}^d \frac{\|W_i\|_F^2}{\sigma_i^2} + \ln \frac{2m}{\delta} \right)}.$$

*where $\gamma_i = \frac{\sigma_i \sqrt{h_i} \sqrt{h_{i-1}}}{\mu^2 \|W_i\|_F}$ and $C_\delta = 2\sqrt{\ln(dh/\delta)}.$*

Figure 5: Verifying the conditions of Theorem 1 on a 10 layer perceptron with 1000 hidden units in each layer, i.e. more than 10,000,000 parameters on MNIST. We have numerically checked that all values are within the displayed range. **Left**: $C1$: condition number of the network, i.e. $\frac{1}{\mu}$. **Middle**: $C2$: the ratio of activations that flip based on magnitude of perturbation. **Right**: $C3$ : the ratio of norm of incoming weights to each hidden units with respect to average of the same quantity over hidden units in the layer.

To understand the above generalization error bound, consider choosing $\gamma_i = \frac{\sigma}{C_\delta d}$, and we get a bound that simplifies as follows:

$$\mathbb{E}_{\boldsymbol{\nu}\sim\mathcal{N}(0,\sigma)^n}[L(f_{\mathbf{w}+\boldsymbol{\nu}})] - \widehat{L}(f_{\mathbf{w}}) \leq O\left(\sigma\left(1 + (1 + \sigma C_2 C_3)C_2\right)C_L\frac{\sum_x \|f_{\mathbf{w}}(x)\|_F}{m}\right)$$
$$+ \sqrt{\frac{1}{m}\left(\frac{d^2}{\mu^4}\sum_{i=1}^{d}\frac{h_i h_{i-1}}{\sigma^2} + \ln\frac{2m}{\delta}\right)}$$

If we choose large $\sigma$, then the network will have higher expected sharpness but smaller 'norm' and vice versa. Now one can optimize over the choice of $\sigma$ to balance between the terms on the right hand side and get a better capacity bound. For any reasonable choice of $\sigma$, the generalization error above, depends only linearly on depth and does not have any exponential dependence, unlike other notions of generalization. Also the error gets worse with decreasing $\mu$ and increasing $C_2, C_3$ as the sharpness of the network increases which is in accordance with our discussion of the conditions above.

Additionally the conditions $C1 - C3$ actually hold for networks trained in practice as we verify in Figure 5, and our experiments suggest that, $\mu \geq 1/4, C2 \leq 5$ and $C3 \leq 3$. Figure 6 compares condition $C1$, $C2$ and $C3$ on learned weights to that of random initialization respectively. Interestingly, we observe that the network with learned weights is very similar to its random initialization in terms of these conditions.

**Proof of Theorem 1** We bound the expectation as follows:

$$\mathbb{E}\left|\widehat{L}(f_{\mathbf{w}+\boldsymbol{\nu}}(x)) - \widehat{L}(f_{\mathbf{w}}(x))\right|$$
$$\leq C_L \mathbb{E}\|f_{\mathbf{w}+\boldsymbol{\nu}}(x) - f_{\mathbf{w}}(x)\|_F$$
$$\stackrel{(i)}{=} C_L \mathbb{E}\|(W+\boldsymbol{\nu})_d\left(\Pi_{i=1}^{d-1}\widehat{D}_i(W+\boldsymbol{\nu})_i\right) * x - W_d\left(\Pi_{i=1}^{d-1}D_i W_i\right) * x\|_F$$
$$\leq C_L \mathbb{E}\|(W+\boldsymbol{\nu})_d\left(\Pi_{i=1}^{d-1}D_i(W+\boldsymbol{\nu})_i\right) * x - W_d\left(\Pi_{i=1}^{d-1}D_i W_i\right) * x\|_F$$
$$\quad + C_L \mathbb{E}\|(W+\boldsymbol{\nu})_d\left(\Pi_{i=1}^{d-1}\widehat{D}_i(W+\boldsymbol{\nu})_i\right) * x - (W+\boldsymbol{\nu})_d\left(\Pi_{i=1}^{d-1}D_i(W+\boldsymbol{\nu})_i\right) * x\|_F$$
$$\leq C_L \mathbb{E}\|(W+\boldsymbol{\nu})_d\left(\Pi_{i=1}^{d-1}D_i(W+\boldsymbol{\nu})_i\right) * x - W_d\left(\Pi_{i=1}^{d-1}D_i W_i\right) * x\|_F + C_L \mathbb{E}\|Err_d\|_F, \quad (7)$$

where $Err_d = \|(W+\boldsymbol{\nu})_d\left(\Pi_{i=1}^{d-1}\widehat{D}_i(W+\boldsymbol{\nu})_i\right) * x - (W+\boldsymbol{\nu})_d\left(\Pi_{i=1}^{d-1}D_i(W+\boldsymbol{\nu})_i\right) * x\|_F$ and $\widehat{D}_i$ is the diagonal matrix with 0's and 1's corresponding to the activation pattern of the perturbed network $f_{\mathbf{w}+\boldsymbol{\nu}}(x)$.

The first term in the equation (7) corresponds to error due to perturbation of a network with unchanged activations (linear network). Intuitively this is small when any subset of successive layers of the network do no interact weakly with each other (not orthogonal to each other). Condition $C1$ captures this intuition and we bound this error in Lemma 8.

(a) Condition $C1$: condition number $\frac{1}{\mu}$ of the network and its decomposition to two cases for learned weights. **Top**: random initialization **Bottom**: learned weights. **Left**: distribution of all combinations of $a \leq c \leq b-1$. **Middle**: when $a < c < b-1$. **Right**: when $c = a$ or $c = b-1$.

(b) Condition $C1$: condition number $\frac{1}{\mu}$ of the network and its decomposition to two cases for random initialization. **Top**: random initialization **Bottom**: learned weights. **Left**: distribution of all combinations of $a \leq c \leq b-1$. **Middle**: when $a < c < b-1$. **Right**: when $c = a$ or $c = b-1$.

(c) Ratio of activations that flip based on the magnitude of perturbation. **Left**: random initialization. **Middle**: learned weights. **Right**: learned weights (zoomed in).

(d) From left to right: Condition $C3$ for random initialization and learned network, output values for random and learned network

Figure 6: Comparing conditions in Theorem 1 on learned weights to that of random initialization. We have trained a 10 layer perceptron with 1000 hidden units in each layer, i.e. more than 10,000,000 parameters on MNIST. We have numerically checked that all values are within the displayed range.

**Lemma 1.** *Let $\nu_i$ be a random $h_i \times h_{i-1}$ matrix with each entry distributed according to $\mathcal{N}(0, \sigma_i^2)$. Then, under the condition $C1$,*

$$\mathbb{E}\|(W+\nu)_d\left(\Pi_{i=1}^{d-1}D_i(W+\nu)_i\right)*x - W_d\left(\Pi_{i=1}^{d-1}D_iW_i\right)*x\|_F$$
$$\leq \left(\Pi_{i=1}^d\left(1 + \frac{\sigma_i\sqrt{h_ih_{i-1}}}{\mu^2\|D_iW_i\|_F}\right) - 1\right)\|f_{\mathbf{w}}(x)\|_F.$$

The second term in the equation (7) captures the perturbation error due to change in activations. If a tiny perturbation can cause exponentially many changes in number of active nodes, then that network will have huge sharpness. Condition $C2$ and $C3$ essentially characterize the behavior of sensitivity of activation patterns to perturbations, leading to a bound on this term in Lemma 2.

**Lemma 2.** *Let $\nu_i$ be a random $h_i \times h_{i-1}$ matrix with each entry distributed according to $\mathcal{N}(0, \sigma_i^2)$. Then, under the conditions $C1$, $C2$ and $C3$, with probability $\geq 1 - \delta$, for all $1 \leq k \leq d$,*

$$\|\widehat{D}_k - D_k\|_1 \leq O\left(C_2h_kC_\delta\sigma_k\|f_{\mathbf{w}}^{k-1}\|_F\right)$$

*and*

$$\mathbb{E}\|Err_k\|_F \leq O\left(\Pi_{i=1}^k\left(1 + \gamma_iC_2C_3\right)\left(\Pi_{i=1}^k(1 + \gamma_iC_\delta C_2) - 1\right)\|f_{\mathbf{w}}^k\|_F\right).$$

*where $\gamma_i = \frac{\sigma_i\sqrt{h_i}\sqrt{h_{i-1}}}{\mu^2\|D_iW_i\|_F}$ and $C_\delta = 2\sqrt{\ln(dh/\delta)}$.*

Hence, from Lemma 8 and Lemma 2 we get,

$$\mathbb{E}\left|\widehat{L}(f_{\mathbf{w}+\nu}(x)) - \widehat{L}(f_{\mathbf{w}}(x))\right|$$
$$\leq \left[\Pi_{i=1}^d\left(1+\gamma_i\right) - 1 + \Pi_{i=1}^d\left(1 + \gamma_iC_2C_3\right)\left(\Pi_{i=1}^d(1 + \gamma_iC_\delta C_2) - 1\right)\right]C_L\|f_{\mathbf{w}}(x)\|_F.$$

Here $\gamma_i = \frac{\sigma_i\sqrt{h_i}\sqrt{h_{i-1}}}{\mu^2\|D_iW_i\|_F}$. Substituting the above bound on expected sharpness in the PAC-Bayes result (equation (5)), gives the result.

*Proof of Lemma 1.* Define $g_{\mathbf{w},\nu,s}(x)$ as the network $f_{\mathbf{w}}$ with weight $W_i$ in every layer $i \in s$ replaced by $\nu_i$. Hence,

$$\|(W+\nu)_d\left(\Pi_{i=1}^{d-1}D_i(W+\nu)_i\right)*x - W_d\left(\Pi_{i=1}^{d-1}D_iW_i\right)*x\|_F$$
$$\leq \|\sum_i g_{\mathbf{w},\nu,\{i\}}(x)\|_F + \|\sum_{i,j} g_{\mathbf{w},\nu,\{i,j\}}(x)\|_F + \cdots + \|f_\nu(x)\|_F \qquad (8)$$

**Base case:** First we show the bound for terms with one noisy layer. Let $g_{\mathbf{w},\nu,\{k\}}(x)$ denote $f_{\mathbf{w}}(x)$ with weights in layer $k$, $W_k$ replaced by $\nu_k$. Now notice that,

$$\mathbb{E}\|g_{\mathbf{w},\nu,\{k\}}(x)\|_F = \mathbb{E}\|W_d\Pi_{i=k+1}^{d-1}D_iW_i*D_k\nu_k*\left(\Pi_{i=1}^{k-1}D_iW_i\right)*x\|_F$$
$$\overset{(i)}{\leq} \sigma_k\|W_d\Pi_{i=k+1}^{d-1}D_iW_i\|_F\|\|\left(\Pi_{i=1}^{k-1}D_iW_i\right)*x\|_F$$
$$\overset{(ii)}{\leq} \sigma_k\frac{\sqrt{h_kh_{k-1}}}{\mu^2\|D_kW_k\|_F}\|W_d\left(\Pi_{i=1}^{d-1}D_iW_i\right)*x\|_F$$
$$= \sigma_k\frac{\sqrt{h_kh_{k-1}}}{\mu^2\|D_kW_k\|_F}\|f_{\mathbf{w}}(x)\|_F.$$

$(i)$ follows from Lemma 3. $(ii)$ follows from condition $C1$.

**Induction step:** Let for any set $s \subset [d], |s| = k$, the following holds:

$$\mathbb{E}\|g_{\mathbf{w},\nu,s}(x)\|_F \leq \|f_{\mathbf{w}}(x)\|_F\Pi_{i\in s}\sigma_i\frac{\sqrt{h_ih_{i-1}}}{\mu^2\|D_iW_i\|_F}.$$

We will prove this now for terms with $k + 1$ noisy layers.

$$\mathbb{E}\|g_{\mathbf{w},\boldsymbol{\nu},s\cup\{j\}},x)\|_F \leq \sigma_j \frac{\sqrt{h_j h_{j-1}}}{\mu^2 \|D_j W_j\|} \mathbb{E}\|g_{\mathbf{w},\boldsymbol{\nu},s}(x)\|_F$$

$$\leq \sigma_j \frac{\sqrt{h_j h_{j-1}}}{\mu^2 \|D_j W_j\|} \|f_{\mathbf{w}}(x)\|_F \Pi_{i\in s} \sigma_i \frac{\sqrt{h_i h_{i-1}}}{\mu^2 \|D_i W_i\|_F}$$

$$= \|f_{\mathbf{w}}(x)\|_F \Pi_{i\in s\cup\{j\}} \sigma_i \frac{\sqrt{h_i h_{i-1}}}{\mu^2 \|D_i W_i\|_F}$$

Substituting the above expression in equation (8) gives,

$$\|(W+\boldsymbol{\nu})_d \left(\Pi_{i=1}^{d-1} D_i (W+\boldsymbol{\nu})_i\right) * x - W_d \left(\Pi_{i=1}^{d-1} D_i W_i\right) * x\|_F$$

$$\leq \left(\Pi_{i=1}^d \left(1 + \frac{\sigma_i \sqrt{h_i h_{i-1}}}{\mu^2 \|D_i W_i\|_F}\right) - 1\right) \|f_{\mathbf{w}}(x)\|_F.$$

$\square$

*Proof of Lemma 2.* We prove this lemma by induction on $k$. Recall that $\widehat{D}_i$ is the diagonal matrix with 0's and 1's corresponding to the activation pattern of the perturbed network $f_{\mathbf{w}+\boldsymbol{\nu}}(x)$. Let $C_\delta = 2\sqrt{\ln(dh/\delta)}$ and $1_E$ denote the indicator function, that is 1 if the event $E$ is true, 0 else. We also use $f_{\mathbf{w}}^k(x)$ to denote the network truncated to level $k$, in particular $f_{\mathbf{w}}^k(x) = \Pi_{i=1}^k D_k W_k x$.

**Base case:**

$$\|\widehat{D}_1 - D_1\|_1 = \sum_i 1_{\langle(W+\boldsymbol{\nu})_{1,i},x\rangle * \langle W_{1,i},x\rangle < 0} = \sum_i 1_{\langle(\mathbf{w})_{1,i},x\rangle^2 < -\langle(\boldsymbol{\nu})_{1,i},x\rangle * \langle(\mathbf{w})_{1,i},x\rangle}$$

$$\leq \sum_i 1_{|\langle(\mathbf{w})_{1,i},x\rangle| < |\langle(\boldsymbol{\nu})_{1,i},x\rangle|}.$$

Since $\boldsymbol{\nu}_1$ is a random Gaussian matrix, and $\|x\| \leq 1$, for any $i$, $|\langle(\boldsymbol{\nu})_{1,i},x\rangle| \leq 2\sigma_1\sqrt{\ln(dh/\delta)} = \sigma_1 C_\delta$ with probability greater than $1 - \frac{\delta}{d}$. Hence, with probability $\geq 1 - \frac{\delta}{d}$,

$$\|\widehat{D}_1 - D_1\|_1 \leq \sum_i 1_{|\langle(\mathbf{w})_{1,i},x\rangle| \leq \sigma_1 C_\delta} \leq C_2 h_1 \sigma_1 C_\delta.$$

This completes the base case for $k = 1$. $\widehat{D}_1$ is a random variable that depends on $\boldsymbol{\nu}_1$. Hence, in the remainder of the proof, to avoid this dependence, we separately bound $\widehat{D}_1 - D$ using the expression above and compute expectation only with respect to $\boldsymbol{\nu}_1$. With probability $\geq 1 - \frac{\delta}{d}$,

$$\mathbb{E}\|Err_1\|_F = \mathbb{E}\|\widehat{D}_1 * (W+\boldsymbol{\nu})_1 x - D_1 * (W+\boldsymbol{\nu})_1 x\|_F$$

$$\leq \mathbb{E}\|(\widehat{D}_1 - D_1) * W_1 x\|_F + \mathbb{E}\|(\widehat{D}_1 - D_1) * \boldsymbol{\nu}_1 x\|_F$$

$$\overset{(i)}{\leq} \sqrt{C_2 h_1 \sigma_1 C_\delta} \sigma_1 + \sqrt{C_2 h_1 \sigma_1 C_\delta} \sigma_1$$

$$= 2\sqrt{C_2 h_1 \sigma_1 C_\delta} \sigma_1.$$

$(i)$ follows because, each hidden node in $\mathbb{E}\|(\widehat{D}_1 - D_1) * W_1 x\|_F$ has norm less than $\sigma_1 C_\delta$ (as it changed its activation), number of such units is less than $C_2 h_1 \sigma_1 C_\delta$.

$k = 1$ case does not capture all the intricacies and dependencies of higher layer networks. Hence we also evaluate the bounds for $k = 2$.

$$\|\widehat{D}_2 - D_2\|_1 \leq \sum_i 1_{\langle(W+\boldsymbol{\nu})_{2,i},f_{\mathbf{w}+\boldsymbol{\nu}}^1\rangle * \langle W_{2,i},f_{\mathbf{w}}^1\rangle \leq 0} \leq \sum_i 1_{|\langle W_{2,i},f_{\mathbf{w}}^1\rangle| \leq |\langle\boldsymbol{\nu}_{2,i},f_{\mathbf{w}+\boldsymbol{\nu}}^1\rangle| + |\langle W_{2,i},f_{\mathbf{w}+\boldsymbol{\nu}}^1 - f_{\mathbf{w}}^1\rangle|}$$

Now, with probability $\geq 1 - \frac{2\delta}{d}$ we get:

$$|\langle \boldsymbol{\nu}_{2,i}, f^1_{\mathbf{w}+\boldsymbol{\nu}}\rangle| + |\langle W_{2,i}, f^1_{\mathbf{w}+\boldsymbol{\nu}} - f^1_{\mathbf{w}}\rangle|$$

$$\leq C_\delta \sigma_2 \left(\|f^1_{\mathbf{w}}\|_F + 2\sqrt{C_2 h_1 \sigma_1 C_\delta}\sigma_1\right) + \|W_{2,i}\| 2\sqrt{C_2 h_1 \sigma_1 C_\delta}\sigma_1$$

$$\leq C_\delta \sigma_2 \left(\|f^1_{\mathbf{w}}\|_F + 2\sqrt{C_2 h_1 \sigma_1 C_\delta}\sigma_1\right) + C_3 \frac{\|D_2 W_2\|_F}{\sqrt{h_2}} 2\sqrt{C_2 h_1 \sigma_1 C_\delta}\sigma_1$$

$$\overset{(i)}{\leq} C_\delta \sigma_2 \left(\|f^1_{\mathbf{w}}\|_F + 2\sqrt{\frac{\hat{\sigma_1}}{\sqrt{h_i + h_{i-1}}}}\hat{\sigma_1}\right) + 2\hat{\sigma_1}\frac{C_3 \|f_{\mathbf{w}}(x)\|_F^{1/d}}{\mu}\sqrt{\frac{\hat{\sigma_1}}{\sqrt{h_i + h_{i-1}}}}$$

$$= C_\delta \sigma_2 \left(\|f^1_{\mathbf{w}}\|_F + \beta_1 \hat{\sigma_1}\right) + \frac{C_3 \|f_{\mathbf{w}}(x)\|_F^{1/d}}{\mu}\beta_1 \hat{\sigma_1}$$

where, $\beta_i = 2\sqrt{\frac{\hat{\sigma_1}}{\sqrt{h_i + h_{i-1}}}}$. $(i)$ follows from condition $C1$, which results in $\Pi_{i=2}^d \frac{\mu \|D_i W_i\|_F}{\sqrt{h_i}}\frac{\mu \|D_1 W_1 x\|_F}{\sqrt{h_1}} \leq \|f_{\mathbf{w}}(x)\|_F$. Hence, if we consider the rebalanced network[5] where all layers have same values for $\frac{\mu \|D_i W_i\|_F}{\sqrt{h_i}}$, we get, $\frac{\mu \|D_i W_i\|_F}{\sqrt{h_i}} \leq \|f_{\mathbf{w}}(x)\|_F^{1/d}$. Also the above equations follow from setting, $\sigma_i = \frac{\hat{\sigma_i}}{C_2 C_\delta \sqrt{h_i + h_{i-1}}}$.

Hence, with probability $\geq 1 - \frac{2\delta}{d}$,

$$\|\widehat{D}_2 - D_2\|_1 \leq C_2 * h_2 \left(C_\delta \sigma_2 \left(\|f^1_{\mathbf{w}}\|_F + \beta_1 \hat{\sigma_1}\right) + \frac{C_3 \|f_{\mathbf{w}}(x)\|_F^{1/d}}{\mu}\beta_1 \hat{\sigma_1}\right).$$

Since, we choose $\sigma_i$ to scale as some small number $O(\sigma)$, in the above expression the first term scales as $O(\sigma)$ and the last two terms decay at least as $O(\sigma^{3/2})$. Hence we do not include them in the computation of $Err$.

$$\mathbb{E}\|Err_2\|_F = \mathbb{E}\|\widehat{D}_2(W+\boldsymbol{\nu})_2 * \widehat{D}_1 * (W+\boldsymbol{\nu})_1 x - D_2(W+\boldsymbol{\nu})_2 * D_1 * (W+\boldsymbol{\nu})_1 x\|_F$$

$$\leq \mathbb{E}\|(\widehat{D}_2 - D_2)(W+\boldsymbol{\nu})_2 * (\widehat{D}_1 - D_1) * (W+\boldsymbol{\nu})_1 x\|_F + \mathbb{E}\|D_2(W+\boldsymbol{\nu})_2 * (\widehat{D}_1 - D_1) * (W+\boldsymbol{\nu})_1 x\|_F$$

$$+ \mathbb{E}\|(\widehat{D}_2 - D_2)(W+\boldsymbol{\nu})_2 * D_1 * (W+\boldsymbol{\nu})_1 x\|_F.$$

We will bound now the first term in the above expression. With probability $\geq 1 - \frac{2\delta}{d}$,

$$\mathbb{E}\|(\widehat{D}_2 - D_2)(W+\boldsymbol{\nu})_2 * (\widehat{D}_1 - D_1) * (W+\boldsymbol{\nu})_1 x\|_F$$

$$\leq \mathbb{E}\|(\widehat{D}_2 - D_2)W_2 * (\widehat{D}_1 - D_1) * W_1 x\|_F + \mathbb{E}\|(\widehat{D}_2 - D_2)W_2 * (\widehat{D}_1 - D_1) * \boldsymbol{\nu}_1 x\|_F$$

$$+ \mathbb{E}\|(\widehat{D}_2 - D_2)\boldsymbol{\nu}_2 * (\widehat{D}_1 - D_1) * W_1 x\|_F + \mathbb{E}\|(\widehat{D}_2 - D_2)\boldsymbol{\nu}_2 * (\widehat{D}_1 - D_1) * \boldsymbol{\nu}_1 x\|_F$$

$$\leq 2\sqrt{C_2 * h_2 C_\delta \sigma_2 \|f^1_W\|_F} C_\delta \sigma_2 \|f^1_W\|_F \sqrt{C_2 * h_1 * C_\delta \sigma_1} C_\delta \sigma_1$$

$$+ 2\sqrt{C_2 * h_2 C_\delta \sigma_2 \|f^1_W\|_F} C_\delta \sigma_2 \sqrt{h_1}\sqrt{C_2 * h_1 * C_\delta \sigma_1} C_\delta \sigma_1 + O(\sigma^2)$$

$$\leq 4\|f^2_{\mathbf{w}}\|_F \frac{C_\delta^2 \sigma_2 \sigma_1 \sqrt{h_1}}{\mu \|D_2 W_2\|_F}\Pi_{i=1}^2 \sqrt{C_2 h_i C_\delta \sigma_i}.$$

**Induction step:**

Now we assume the statement for all $i \leq k$ and prove it for $k + 1$. $\|\widehat{D}_k - D_k\|_1 \leq O\left(C_2 h_k C_\delta \sigma_k \|f^{k-1}_{\mathbf{w}}\|_F\right)$ and $\mathbb{E}\|Err_k\|_F \leq$

$$O\left(\Pi_{i=1}^{k}\left(1+\frac{\sigma_i\sqrt{h_i}\sqrt{h_{i-1}}C_2C_3}{\mu^2\|W_i\|_F}\right)\left(\Pi_{i=1}^{k}(1+\frac{\sigma_i\sqrt{h_i}\sqrt{h_{i-1}}C_\delta C_2}{\mu^2\|W_i\|_F})-1\right)\|f_{\mathbf{w}}^k\|_F\right).$$ Now we prove the statement for $k+1$.

$$\|\widehat{D}_{k+1}-D_{k+1}\|_1 = \sum_i \mathbb{1}_{\langle(W+\boldsymbol{\nu})_{k+1,i},\Pi_{i=1}^k\widehat{D}_i(W+\boldsymbol{\nu})_i * x\rangle * \langle W_{2,i},D_1 W_1 x\rangle \leq 0}$$

$$\leq \sum_i \mathbb{1}_{|\langle W_{k+1,i},\Pi_{i=1}^k\widehat{D}_i(W+\boldsymbol{\nu})_i * x\rangle| \leq |\langle \boldsymbol{\nu}_{k+1,i},\Pi_{i=1}^k\widehat{D}_i(W+\boldsymbol{\nu})_i * x\rangle|}$$

$$= \sum_i \mathbb{1}_{|\langle W_{k+1,i},f_{\mathbf{w}+\boldsymbol{\nu}}^k\rangle| \leq |\langle \boldsymbol{\nu}_{k+1,i},f_{\mathbf{w}+\boldsymbol{\nu}}^k\rangle|}$$

$$\leq \sum_i \mathbb{1}_{|\langle W_{k+1,i},f_W^k\rangle| \leq |\langle \boldsymbol{\nu}_{k+1,i},f_{\mathbf{w}}^k\rangle| + |\langle \boldsymbol{\nu}_{k+1,i},f_{\mathbf{w}+\boldsymbol{\nu}}^k - f_{\mathbf{w}}^k\rangle| + |\langle W_{k+1,i},f_{\mathbf{w}+\boldsymbol{\nu}}^k - f_{\mathbf{w}}^k\rangle|}$$

Hence, with probability $\geq 1 - \frac{k\delta}{d}$,

$$\|\widehat{D}_{k+1}-D_{k+1}\|_1 \leq C_2 h_{k+1}\left[C_\delta\sigma_{k+1}(\|f_{\mathbf{w}}^k\|_F + \|f_{\mathbf{w}+\boldsymbol{\nu}}^k - f_{\mathbf{w}}^k\|_F) + \|W_{k+1,i}\|\|f_{\mathbf{w}+\boldsymbol{\nu}}^k - f_{\mathbf{w}}^k\|_F\right]$$
$$\leq C_2 h_{k+1}C_\delta\sigma_{k+1}\|f_{\mathbf{w}}^k\|_F + C_2 h_{k+1}C_\delta\sigma_{k+1}\|f_{\mathbf{w}+\boldsymbol{\nu}}^k - f_{\mathbf{w}}^k\|_F + C_2 h_{k+1}\|W_{k+1,i}\|\|f_{\mathbf{w}+\boldsymbol{\nu}}^k - f_{\mathbf{w}}^k\|_F.$$

Now we will show that the last two terms in the above expression scale as $O(\sigma^2)$. For that, first notice that $\|f_{\mathbf{w}+\boldsymbol{\nu}}^k - f_{\mathbf{w}}^k\|_F \leq \left(\Pi_{i=1}^{k}\left(1+\frac{\sigma_i\sqrt{h_i h_{i-1}}}{\mu^2\|D_i W_i\|_F}\right)-1\right)\|f_{\mathbf{w}}(x)\|_F + Err_k$, from lemma 1. Note that the second term in the above expression clearly scale as $O(\sigma^2)$.

Hence,

$$\|\widehat{D}_{k+1}-D_{k+1}\|_1 \leq O\left(C_2 h_{k+1}C_\delta\sigma_{k+1}\|f_{\mathbf{w}}^k\|_F\right).$$

$$\|Err_{k+1}\| = \|f_{\mathbf{w}+\boldsymbol{\nu}}^{k+1} - \tilde{f}_{\mathbf{w}+\boldsymbol{\nu}}^{k+1}\|_F$$
$$= \|\widehat{D}_{k+1}(W+\boldsymbol{\nu})_{k+1}\Pi_{i=1}^{k+1}\widehat{D}_i(W+\boldsymbol{\nu})_i x - D_{k+1}(W+\boldsymbol{\nu})_{k+1}\Pi_{i=1}^{k+1}D_i(W+\boldsymbol{\nu})_i x\|_F$$
$$\leq \|(\widehat{D}_{k+1}-D_{k+1})(W+\boldsymbol{\nu})_{k+1}\Pi_{i=1}^{k+1}D_i(W+\boldsymbol{\nu})_i x\|_F + \|\widehat{D}_{k+1}(W+\boldsymbol{\nu})_{k+1}Err_k\|_F$$
$$\leq \|(\widehat{D}_{k+1}-D_{k+1})(W+\boldsymbol{\nu})_{k+1}\Pi_{i=1}^{k+1}D_i(W+\boldsymbol{\nu})_i x\|_F + \|(\widehat{D}_{k+1}-D_{k+1})(W+\boldsymbol{\nu})_{k+1}Err_k\|_F$$
$$+ \|D_{k+1}(W+\boldsymbol{\nu})_{k+1}Err_k\|_F$$

Substituting the bounds for $\widehat{D}_{k+1}-D_{k+1}$ and $Err_k$ gives us, with probability $\geq 1 - \frac{k\delta}{d}$.

$$\mathbb{E}\|Err_{k+1}\| \leq \sqrt{C_2 h_{k+1}C_\delta\sigma_{k+1}\|f_{\mathbf{w}}^k\|_F}C_\delta\sigma_{k+1}\|f_W^k\|_F\mathbb{E}\|\Pi_{i=1}^{k+1}D_i(W+\boldsymbol{\nu})_i x\|_F$$
$$+ \mathbb{E}\|Err_k\|_F\left(\sqrt{C_2 h_{k+1}C_\delta\sigma_{k+1}\|f_{\mathbf{w}}^k\|_F}C_\delta\sigma_{k+1}\|f_{\mathbf{w}}^k\|_F + \|D_{k+1}W_{k+1}\|_F + \sigma_{k+1}\sqrt{h_{k+1}}\right)$$

Now we bound the above terms following the same approach as in proof of Lemma 1, by considering all possible replacements of $W_i$ with $\boldsymbol{\nu}_i$. That gives us the result.

$\square$

**Lemma 3.** *Let $A$, $B$ be $n_1 \times n_2$ and $n_3 \times n_4$ matrices and $\boldsymbol{\nu}$ be a $n_2 \times n_3$ entrywise random Gaussian matrix with $\boldsymbol{\nu}_{ij} \sim \mathcal{N}(0,\sigma)$. Then,*

$$\mathbb{E}\left[\|A * \boldsymbol{\nu} * B\|_F\right] \leq \sigma\|A\|_F\|B\|_F.$$

*Proof.* By Jensen's inequality,

$$\mathbb{E}\left[\|A * \boldsymbol{\nu} * B\|_F\right]^2 \leq \mathbb{E}\left[\|A * \boldsymbol{\nu} * B\|_F^2\right]$$

$$= \mathbb{E}\left[\left(\sum_{ij}\sum_{kl} A_{ik}\boldsymbol{\nu}_{kl}B_{lj}\right)^2\right]$$

$$= \sum_{ij}\sum_{kl} A_{ik}^2 \mathbb{E}\left[\boldsymbol{\nu}_{kl}^2\right] B_{lj}^2$$

$$= \sigma^2 \|A\|_F^2 \|B\|_F^2.$$

$\square$

## Footnotes

[3]Xu and Mannor [29] have defined the robustness as a property of learning algorithm given the model class and the training set. Here since we are focused on the learned model, we introduce it as a property of the model.

[4]Similar to margin-based bounds, we drop the term that depends on the diameter of the input space.

[5]The parameters of ReLu networks can be scaled between layers without changing the function