[Reviews · NeurIPS 2017]

Reviewer 1



This paper presents an initial empirical analysis of different metrics which can be used to predict the generalization error of a neural network. These metrics include sharpness, lipschitz continuity and various norms on the weight matrices. Finally they suggest three different conditions to prevent pathological solutions by bounding the sharpness of solution. While the analysis presented in the paper is inconclusive, these intermediate results are of interest. Minor details : There are a few grammatical mistakes and typos in the paper. e.g. Ln 103 : we check weather -> we check whether Ln 141 - This suggest -> This suggests

Reviewer 2



Update after rebuttal: quote the rebuttal "Even this very simple variability in architecture, proves challenging to study using the complexity measures suggested. We certainly intend to study also other architectural differences." I hope the authors include the discussions (and hopefully some experiment results) about applying the proposed techniques to the comparison of models with different architectures in the final version if it get accepted. The current results are already useful steps towards understanding deep neural networks by their own, but having this results is really a great add on to this paper. ------------------------- This paper investigated primarily two (related) questions: what is the complexity measure that is implicitly enforced to allow good generalization performance of large deep neural networks? Given different global minimizers of the empirical risk, can we tell which one generalize better based on the complexity measure. Although no definite answer is given in this paper, a few options are explored and some of them seem to be positive choice. More specifically, norm based complexity measure and (average) sharpness based complexity measures are empirically investigated. The sharpness based measure are further justified via PAC-Bayes bound on generalization. There is no discussion on how or why those complexity measures are enforced during training of neural networks, but the explorations are interesting by themselves. Although a theorem is given to control the sharpness of a network under a number of assumptions (C1-C3), this is primarily an empirical paper. Therefore, the reviewer thinks it would be more useful to include more extensive experiments. More specifically, could the proposed measures be useful when comparing across different architectures? Could it be useful when compare to different global minimizers from different algorithms (variants of SGD, or SGD with small / large batches)? etc.

Reviewer 3



Summary: Paper studies how one can measure how well a network will generalize. (This is linked to implicit regularization present in our current optimization methods.) In particular, the paper links the capacity of neural networks to four different path-norm measures of the weights of the network. They revisit/define margins, lipschitz continuity and (expected) sharpness as measures of network capacity/generalization. An interesting experiment is done where the ability of these measures to predict how well a network can generalize is carried out: a network is trained on a subset of a training set, along with different fractions of corrupted labels for other datapoints (different fractions for different networks). The measures are then used to predict how well a network might generalize. An interesting paper. (Some of the text has minor typos though.) But it seems that the conclusion from the middle pane of Figures 3, 4 that spectral norm and path l2 norm are the best measures for generalization, as they seem to correlate with test error in both cases, whereas the others don't. (The path l1 norm also does badly). For better comparison, it would have been good to see Figure 4 for CIFAR10 also. Minor comments: I assume the epsilon is added after the softmax (in (2))? Is there some rescaling then? Why does the left of Figure 2 show that sharpness is unstable? This seems to be more of a feature of Figures 3, 4?